**Data Availability Statement:** Raw data of the impact evaluation were generated at the national

# Impact of the extension of a performance-based financing scheme to nutrition services in Burundi on malnutrition prevention and management among children below five: A cluster-randomized control trial

Catherine Korachais[1]*, Sandra Nkurunziza[2,3], Manassé Nimpagaritse[1,4,5], Bruno Meessen[1]

**1** Public Health Department, Institute of Tropical Medicine, Antwerp, Belgium, **2** Global Health Institute, University of Antwerp, Belgium, **3** Health Community Department, University of Burundi, Bujumbura, Burundi, **4** Institut National de Santé Publique, Bujumbura, Burundi, **5** Institut de Recherche Santé et Société, Université Catholique de Louvain, Brussels, Belgium

* Catherine.korachais@protonmail.com

## Abstract

Malnutrition is a huge problem in Burundi. In order to improve the health system response, the Ministry of Health piloted the introduction of malnutrition prevention and care indicators within its performance-based financing (PBF) scheme. Paying for units of services and for qualitative indicators is expected to enhance provision and quality of these nutrition services. The objective of this study is to assess the impacts of this intervention, on both child acute malnutrition recovery rates at health centre level and prevalence of chronic and acute malnutrition among children at community level. This study follows a cluster-randomized controlled evaluation design: 90 health centres (HC) were randomly selected for the study, 45 of them were randomly assigned to the intervention and received payment related to their performance in malnutrition activities, while the other 45 constituted the control group and got a simple budget allocation. Data were collected from baseline and follow-up surveys of the 90 health centres and 6,480 households with children aged 6 to 23 months. From the respectively 1,067 and 1,402 moderate and severe acute malnutrition transcribed files and registers, findings suggest that the intervention had a positive impact on moderate acute malnutrition recovery rates (OR: 5.59, p = 0.039 –at the endline, 78% in the control group and 97% in the intervention group) but not on uncomplicated severe acute malnutrition recovery rate (OR: 1.16, p = 0.751 –at the endline, 93% in the control group and 92% in the intervention group). The intervention also had a significant increasing impact on the number of children treated for acute malnutrition. Analyses from the anthropometric data collected among 12,679 children aged 6–23 months suggest improvements at health centre level did not translate into better results at community level: prevalence of both acute and chronic malnutrition remained high, precisely at the endline, acute and chronic malnutrition prevalence were resp. 8.80% and 49.90% in the control group and 8.70% and 52.0% in the intervention group, the differences being non-significant. PBF can contribute to a better

institute of public health of Burundi, INSP (https://insp.bi/), for the health facility data, and at the national institute of statistics of Burundi, ISTEEBU (http://www.isteebu.bi/), for the household data. All data are owned by ISTEEBU and are available from them on request, for researchers who meet the criteria for access to confidential data, by sending an e-mail to the Director General of ISTEEBU: isteebu@isteebu.bi or isteebubdi@gmail.com. A synthetic dataset relevant to the analysis developed in the manuscript is provided as a supporting information file (S1 Dataset).

**Funding:** This study was funded by the Health Results Innovation Trust Fund (HRITF) World Bank (https://www.rbfhealth.org/), World Bank contract no. 7266169. The funders had no role in study design, data collection and analysis, decision to publish, or preparation of the manuscript.

**Competing interests:** CK, SN and MN declare no conflict of interest. BM has contributed to the emergence of PBF as a global health policy, through technical assistance, research and knowledge management. He is the lead facilitator of the PBF Community of Practice. He holds minority shares in Blue Square, a firm developing health system software solutions. This does not alter our adherence to PLOS ONE policies on sharing data and materials.

management of malnutrition at HC level; yet, to address the huge problem of child malnutrition in Burundi, additional strategies are urgently required.

## Introduction

The latest *State of Food Security and Nutrition in the World* report signals a rise in world hunger and food insecurity and a reversal of trends after a prolonged decline [1]. Causes of this increase are multiple, with climate variability and extremes identified as among the key drivers. Urgent and considerable action is needed if we want to achieve the SDG goals on food security and nutrition (by 2030, end all forms of malnutrition). Malnutrition rates in many low-income countries (LICs) remain terrible. Stunting is declining too slowly while wasting still impacts the lives of far too many young children. Poor nutrition in the first 1,000 days of a child's life can also lead to stunted growth, which is associated with impaired cognitive ability and later on reduced school and work performance [2].

The state of nutrition in Burundi is particularly alarming. Burundi is a small landlocked country in Eastern Central Africa, with a very high and increasing population density (more than 420 people per sq. km of land area in 2017 [3]). This country is in a fragile situation, being among the poorest in the world in terms of income per capita, and regularly affected by political crises. In 2016/2017, 55.9% of children under five years old suffered from chronic malnutrition (which places the country in the world top 3) and 5.1% from acute malnutrition [4]; these figures were respectively 57.7% and 5.8% in 2010, showing almost no improvement [5].

Malnutrition is multifactorial: poverty and food security are central elements in low-income countries, while water, sanitation and hygiene are found to be important factors of chronic malnutrition among children [6]. There is also a consensus that the health system has a role to play [7], especially at the health centre level, where basic perinatal care, paediatric prevention and care services are provided: generic health services can prevent and manage child health issues that could lead otherwise to malnutrition, while specific to malnutrition interventions aim to prevent, identify and manage malnutrition. In Burundi, a national program is dedicated to the fight against malnutrition [8, 9]. It has adopted and rolled out different strategies promoted by international agencies, including the systematic screening and outpatient treatment for moderate and non-complicated severe acute malnutrition (MAM and SAM) children at health centre (HC) level. Community-based management of acute malnutrition (CMAM), although highly relevant [7], was not sufficiently invested in Burundi at the time of the study. CMAM includes 1) community outreach as the basis for early identification and referral of malnourished children, follow up through home visits, sensitisation and mobilisation of the community; 2) management of MAM; 3) outpatient treatment for children with uncomplicated SAM; and, 4) inpatient treatment for children with complicated SAM. In Burundi, active screening was to be performed by CHWs, and systematic screening at health centres through pre-school consultations and curative consultations of children under five; outpatient treatment services for MAM and uncomplicated SAM cases are integrated in the primary health care services, in about one third of health centres. Complicated SAM cases are managed in the hospital, before being transferred back to outpatient treatment and follow-up at the health centre. As for sensitisation at community level, this was mainly performed by NGOs, in only a few provinces.

Over the last decade, Burundi has taken several bold measures to boost the performance of its health system. One of them has been a nationwide performance-based financing (PBF) policy. Since 2010, a part of health facilities revenue is linked to their performance (measured as a combination of volume and quality of health services). There is evidence that this strategy had positive effects on child and maternal health services in Burundi; studies found that PBF was associated with an increase in the health workforce as well as in health service utilization (incl. vaccination, assisted deliveries, antenatal care) and quality of care [10–13].

In 2012, the Ministry of Health and the World Bank decided to pilot the extension of the PBF system to nutrition services, as a measure to reinforce the recently developed malnutrition prevention and management program [9].

As a strategy, PBF has expanded rapidly in low- and middle-income countries in the last decade. This has been accompanied by a large research program to assess the effectiveness of the strategy across settings and health conditions. Only three studies examined the impact of PBF on any nutrition outcome, with mixed results. Following the Cochrane review's general recommendations for solid evaluations of PBF schemes [14, 15], it was decided that the extension of the PBF scheme to nutrition (the first ever to our knowledge) would be subject to an impact evaluation.

This paper reports findings from this evaluation, with a focus on the two main research questions: (1) did the extension of PBF to nutrition (hereafter, referred as PBF-N) contributed to improve nutritional health services for children below five years old at HC level and; (2) did it reduce the prevalence of malnutrition among children below two years old at community level?

## Methods

### Study design

This interventional study followed a cluster randomised controlled trial (cRCT) design. The intervention design and implementation were under the responsibility of the Ministry of Health. Data collection was carried out by two national survey institutes (the *Institut National de Santé Publique* (INSP) for the facility surveys and the *Institut de statistiques et d'Etudes Economiques du Burundi (ISTEEBU)* for the household surveys). The general coordination of the research was under the responsibility of the Institute of Tropical Medicine of Antwerp, with logistical assistance from Blue Square Bujumbura. The intervention covered all provinces of Burundi, but the urban province of Bujumbura, the capital city. Among the 590 health centres in the country, 193 health centres provided nutrition services in 2014, and among them, 90 were randomly selected for the study. In the intervention group (concerning the 45 randomly allocated HCs), hospitals, HCs and the related community health workers (CHWs) were subject to PBF-N, for two years. In the control group, the 45 HCs received, monthly, a simple budget allocation while hospitals were also subject to PBF-N.

The study protocol was reviewed and approved by ethics committees of the Institute of Tropical Medicine of Antwerp, the University of Antwerp, and the National Ethics Committee of Burundi (see S3–S6 Files). This study is registered with ClinicalTrials.gov, number NCT027211160 [16], and the research protocol has been published elsewhere [17]. Informed written consents were obtained for all participants of the study, that is: health workers, household heads, and mothers (or child caregivers) for themselves and their children.

The CONSORT flow diagram (Fig 1 below) displays the progress of enrolment and intervention allocation of health centres, as well as of follow-up and data analysis along the study. The CONSORT checklist can be found in S1 File.

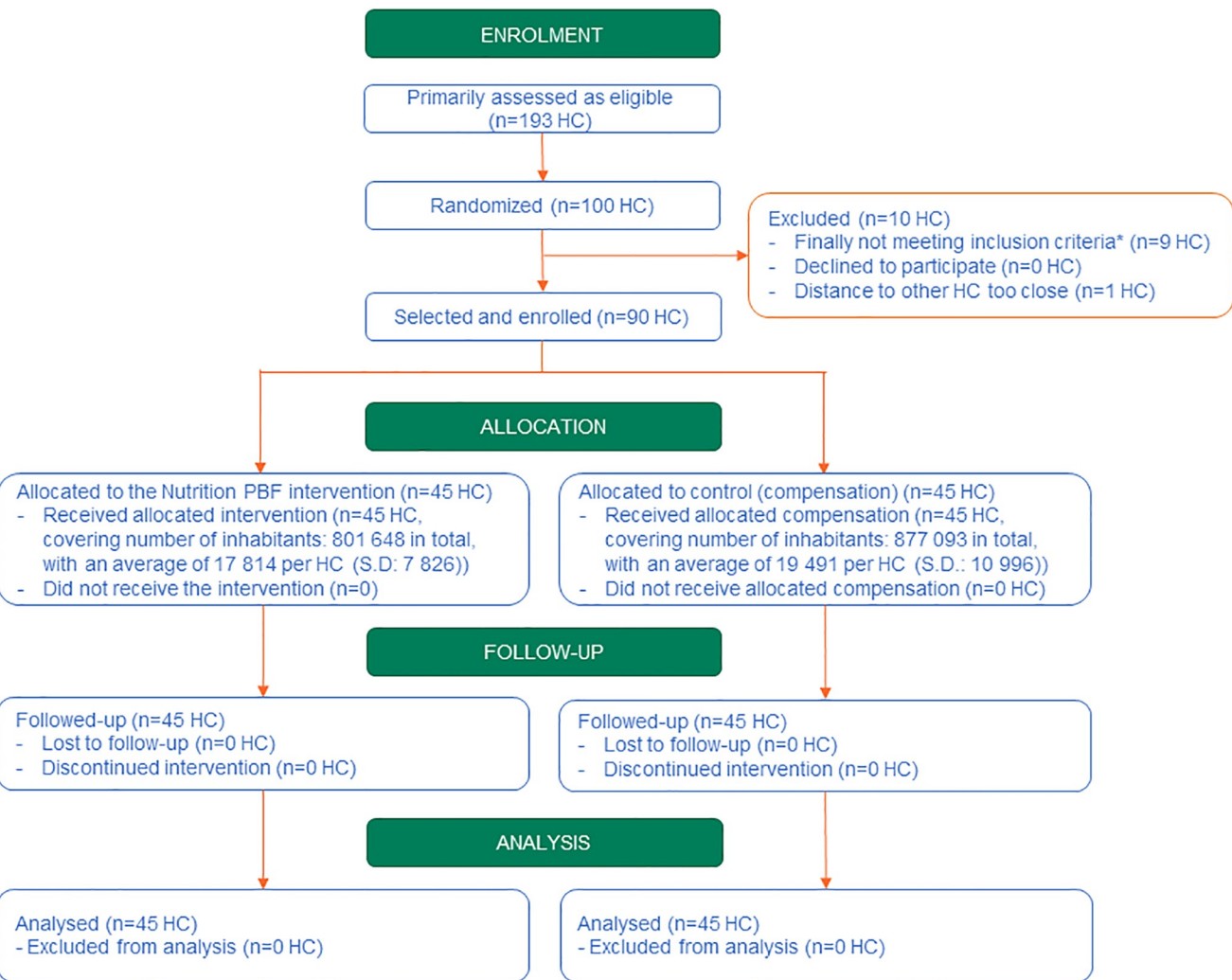

**Fig 1. CONSORT flow diagram.** HC means health centre, S.D. means standard deviation; *Information about the existence of both MAM and SAM services was initially obtained from the Ministry of Health; once the randomization was performed, while gathering more information from each HC, we realised than nine of them did not provide either the MAM (one) or the SAM (eight) services. Providing both services was an inclusion criterion, so these nine HC were removed from the list. Source: authors.

## Participants

HCs were eligible to participate in the study if, at the time of recruitment, i.e. Summer 2014, they provided both moderate and severe acute malnutrition management services, they were already subject to PBF, they were in a rural province, and they were separated from another participating health centre by at least 5 km. Surveyed households, health workers and patients were recruited on the field, at the time of the survey (see the procedures below).

## Randomisation and masking

In order to study rather similar groups of HCs, the 90 participating HCs were paired on various parameters before being allocated to either the intervention or the control groups. Based on data collected during the baseline survey, we first assigned HCs to four different groups, according to the continuation (or not) of MAM services despite the treatment supply

discontinuation, and to their volume of activity (above or below the median). Then in each group, we paired HCs according to their performance in acute malnutrition services, using the acute malnutrition recovery rates. For each pair, the random allocation to the intervention or the control group took place in a public session with HC managers one month before the start of the intervention.

The sample size was computed on the smallest difference in the main outcome that could be considered of public health significance, i.e. a reduction of $\cong$ 25% in acute malnutrition prevalence (2.5% points in absolute terms) in the communities around the intervention HCs as compared to control HCs. Assuming that the intervention would result in decreasing the prevalence of acute malnutrition in children aged 6–23 months from 10% to 7.5% (DHS survey 2010), and assuming that 65 children aged 6–23 months would be surveyed in the catchment area of each health centre, 90 health centres needed to be randomized to either the intervention or control group, for an $\alpha$-error of 5% and a $\beta$-error of 20%, and taking a conservative intraclass correlation (ICC) of 0.25. This was calculated with the formula of [18]. The number of children per HC was increased to 72 to allow for missing or incomplete data, amounting to a total of 6,480 children aged 6–23 months over the 90 selected HCs, at each survey round (baseline in 2014 and endline in 2016). Now, according to the baseline data, the prevalence of acute malnutrition in children aged 6–23 months is 6%, instead of 10%, with an ICC of 0.007. Considering this, the explanatory power for a reduction of 25% (from 6% to 4.5%) was reduced from 80% to 60%, while the explanatory power for a reduction from 6% to 4% remained at 80%.

## Intervention

**Interventions intended for each group.** In the intervention group, the PBF-N intervention consisted in paying for improved performance at three levels—hospital, HC and community health workers (CHWs)–for malnutrition screening, growth promotion as well as acute malnutrition management interventions, all focusing on children below five years old (cf. Table 1 and Fig 2).

A quality of care checklist, with 19 indicators specific to nutrition, was added to the existing one, and was considered for the calculation of payment to facilities (this checklist can be found in the supporting information, S2 File). The whole reliability of the payment system was ensured through a verification system, itself subject to a regular systemic audit [19]. HC and hospital managers as well as controllers received a training regarding the PBF-N, during a workshop that took place in Bujumbura in December 2014, one month before the start of the intervention (January 2015). The intervention at community level was delayed and CHWs

**Table 1. Measures used for the PBF-N payments.**

| Level | Payments are given for: |
|---|---|
| Hospital | (1) success in managing and rehabilitating severe acute malnutrition cases with medical complications and<br>(2) length of stay (number of days) at the hospital due to malnutrition. |
| Health Centre | (1) screening children for malnutrition and caring (or referring if services not available) for rehabilitating acute malnutrition cases, and<br>(2) growth monitoring and promotion activities. |
| Community | (1) screening and referring acute malnutrition cases to health centres,<br>(2) organising classes promoting good food and nutrition behaviours, and<br>(3) organising cooking classes. |

Source: Ministry of Health technical note [20].

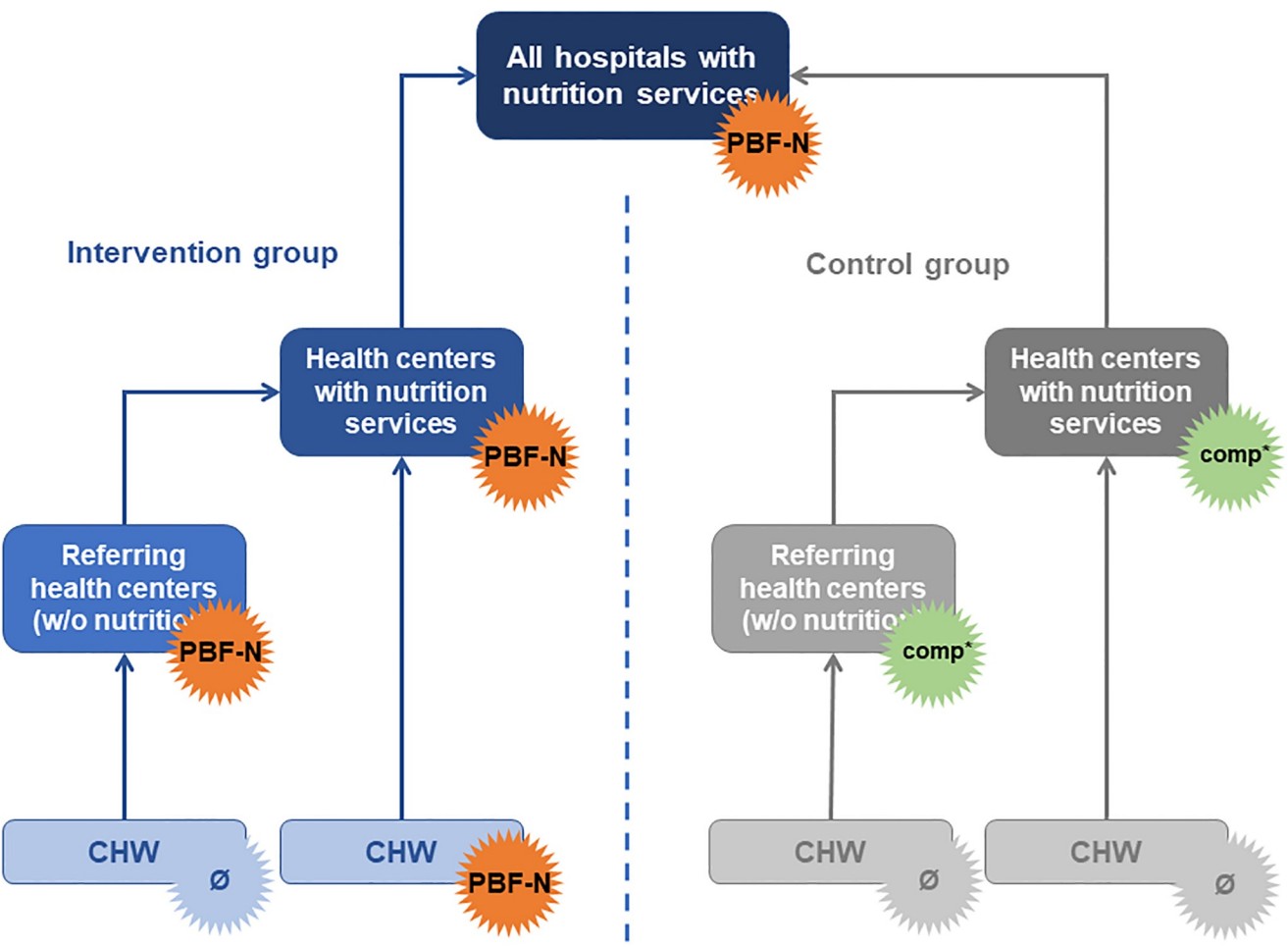

**Fig 2. PBF-N system.** Source: authors. Note: 'comp.' refers to the 'compensation intervention'.

received a training in September 2015, just before the intervention officially started in October 2015. At all levels, the intervention lasted until December 2016, i.e. two years at HC and hospital levels and fifteen months at CHW level.

PBF-N subsidies were transferred to the bank accounts of hospitals, HCs and CHW associations. Revenues from the PBF-N were fungible with other revenues raised by the hospitals and HC (the national PBF scheme, user fees, etc.); their use is under the responsibility of the facility staff who based their allocation decisions on a management tool prioritising purchase of key inputs (e.g. medicines); surplus (if any) is distributed to staff as bonuses, again using a specific management tool. As for CHW associations, one part of the funds could be invested into revenue-generating activities and the rest was shared among CHWs.

In the control group, the hospitals were also subject to the PBF-N, i.e. they received subsidies according to their performance with complicated SAM management (Table 1). HCs were subject to a 'compensation intervention': they received a financial amount equivalent to the PBF subsidies received in the intervention group (weighted average, considering volume of activities and staff). The purpose of this payment was to assure that any future significant difference between the intervention and control groups could be attributed to the payment-for-result rule and not to the extra money injected in the HCs. The CHWs were not subject to any intervention, i.e. they received nothing more than usual.

The initial theory of change of the intervention is described in details in the research proto-col [17]. To sum it up, the introduction of nutrition activities into the PBF program translates policy makers' belief that PBF can trigger some positive changes in the performance of the health personnel, facilities or system which will eventually impact on households and children. To reap the extra income from the PBF-N, service providers (CHWs, HCs and hospitals) are expected to take proactive steps to improve the delivery of nutrition services. In addition, through the PBF-N contract, staff has a clearer view on what performance should be, as far as nutrition services are concerned. In addition, verifications, essential to enforce any PBF inter-vention, may result in more systematic supervisions and qualitative feedbacks, and guide the health workers decisions towards 'performance'. A chain of expected results can be found in S1 Fig. In another paper, recently published, we report on more extensively our findings as for the theory of change [21]. Qualitative data suggest that the most influencing mechanisms related to PBF-N were incentives as well as information provided in the contract and during the initial training (stating clearly the service performance indicators).

## Procedures

**Cross-sectional household surveys.**   Cross-sectional household surveys were performed in all clusters at baseline, in December 2014 and January 2015, and at end-line, in March and April 2017 (i.e. about two years after the PBF-N intervention had started). Surveyed house-holds were recruited at the time of the survey. Eligible households were those living in the area of the study health centres and having a member aged 6–23 months at the time of the survey. As described in the randomization subsection above, the number of children per cluster to be surveyed at community level was set to 72. Around each HC, twelve sub-hills were randomly selected among the sub-hills covering the HC catchment area, and in each sub-hill, six eligible households were selected as follows: a random direction from the centre of the sub-hill was selected and following it led to the first contact with a household, which in turn indicated the next households with children aged 6–23 months. Demographic details about households and children characteristics at baseline, in 2014, are displayed in S1 Table. Chronic malnutrition indicators computed for our sample of children were similar to those found in the nationally representative DHS 2010 survey for same age children, which make us confident that our sam-ple can be considered representative of all households with a toddler living in Burundi at that time.

The nutritional status of the child was assessed through anthropometric measurements; health care behaviour and diet of the child were assessed through a questionnaire administered to the mother (or caregiver). The questions in these sections were a combination of those adapted from the WHO guidance on assessing infant and young child feeding practices [22] as well as those designed by the Child Health and Nutrition Unit of the ITM. Socio-economic characteristics and food security of the household were assessed through a questionnaire administered to the head of the household. The questions used for the assessment of household food security were based on the 2007 Household Food Insecurity Access Scale (HFIAS) Generic Questions, created by the Food and Nutrition Technical Assistance (FANTA) project [23].

Before each survey, fieldworkers received a one-week training. For each survey, 72 survey-ors were deployed across clusters in teams of 12 surveyors (working in duos). Each team was managed by a supervisor. Interviews were performed using Android smartphones with Open Data Kit® software. Data were transferred every evening to the ONA internet data manage-ment platform and checked regularly by two data managers for consistency and completeness. The electronic data entry has the advantage of reducing risks of errors in recording the answers

(thanks to automatic validity checks) and eliminating the need for double data entry from the paper to software transcription and to decrease considerably the time for transcription. Lot quality assessment surveys were also performed by the field coordinator to assess accuracy of anthropometric data in the records.

**Health centre surveys.** The 90 participating HCs were surveyed at baseline, in September-October 2014 and at endline in March-April 2017. Various survey tools were administered at the HC level, in 2014 and 2017. Surveyed health workers and patients were recruited at the time of the surveys, as described below.

To get information on malnutrition recovery rates, in each study HC, a total of 24 individual clinical files were to be transcribed, totalling 1,080 MAM and 1,080 SAM clinical files for each survey wave. This sample size was computed on the smallest difference in the main outcome that can be considered of public health significance in intervention health centres, and assuming that the intervention would result in increasing the recovery rate of both MAM and SAM in children from 80% to 90%, for an $\alpha$-error of 5%, a power of 80% and an ICC of 0.15 [18]. These 24 clinical files were randomly selected among all the closed files of all children enrolled acute malnutrition services (12 files for the MAM care program and 12 files for the SAM care program) during a six-month period preceding the survey where both services were procured with malnutrition treatments (March-September 2013 for baseline and April-October 2016 for endline). In addition, organizational aspects of the HCs as well as of the nutrition services were collected through interviews of managers.

To assess the quality of services, we combined two techniques: patient-provider observation carried out on six paediatric consultations (performed by a maximum of two HWs) and exit interviews at the end of each of these observed consultations, in order to get information on the satisfaction level as well as to record anthropometrics of the children (540 expected observations per survey). For the latter, two health workers were randomly selected among those performing paediatric curative consultations the day of the survey; then six children aged 6–59 months, coming for a consultation the day of the survey, were randomly selected, among the eligible patients in the waiting room, for being observed in consultation and surveyed afterwards. Finally, to assess knowledge of the observed HWs, we used vignettes to measure the practical knowledge on different tasks to perform: a pattern of consultation was proposed and the health worker could ask all the questions (related to history and physical exams) necessary to arrive at a diagnosis and propose a treatment. Three vignettes were administered to every HW observed in consultation.

The 90 HC selected for the study cover all areas of Burundi and all provinces but Bujumbura city have at least one HC in the sample. Although this sample is not representative of the whole health sector in Burundi, since neither hospitals nor community health workers were surveyed, we are confident that this sample is representative of all health centres of the rural part of Burundi. Health centre characteristics and nutrition services performance indicators were rather similar to other nationally representative data.

**PBF Routine heath centre data.** In addition, routine data from the PBF-N program was obtained from the PBF technical unit of the Ministry of Health for the intervention HCs over 2015 and 2016. It provided regular information about the 45 HCs, as for their quantitative and qualitative 'nutrition performance' and the related payments.

## Outcomes

The primary outcomes to test were the prevalence of chronic and acute malnutrition measured at the community level, as well as the MAM and SAM recovery rates among children below five years old at the HC level. At the community level, for statistical power issues, the focus was

made on the age group with the highest prevalence, where we could expect the largest impact, that is, children aged 6–23 months.

Chronic malnutrition identification is based on length-for-age and height-for-age z-scores (LAZ and HAZ) respectively for children below and above two years old. Length-for-age and height-for-age deficits indicate past or chronic inadequacies of nutrition and/or chronic or frequent illness [24]. A child below two years old is stunted when their LAZ is lower than minus two standard deviations (SD) and severely stunted when it's lower than minus three SDs. Acute malnutrition identification, for children below two, is based on weight-for-length z-scores (WLZ), mid-upper arm circumference (MUAC) and the presence of oedema. Severe acute malnutrition is diagnosed either when the WLZ is below minus three SDs or when the MUAC is below 115mm or when oedema is present. MAM is diagnosed when there is no oedema and either the WLZ is between -3SD and -2SD or MUAC is between 115mm and 125mm.

MAM and SAM recovery rates among children below five years old and managed at the HC are based on the transcribed clinical files collected from both MAM and SAM management services. HWs report various elements on these files, including the reason for the end of treatment, which may be recovery, transfer, desertion, non-responding, death, or any other reason.

As for secondary outcomes, at community level we also analysed WLZ, LAZ and MUAC, and at HC level, treatment duration among the MAM and SAM cases followed up and recovered at the HC.

As for intermediary outcomes, we focused on prevention and screening of malnutrition. At community level, we checked the extent to which acute malnutrition cases diagnosed among all surveyed children were indeed followed-up at the HC. At HC level, we looked at the quality of diagnosis during paediatric curative consultations, by (1) administrating fictive cases with three vignettes with three different cases of acute malnutrition (MAM, non-complicated and complicated SAM), and (2) by observing real consultations (combined with anthropometric measurements taken during an exit interview): was acute malnutrition diagnosed during consultation, and, if so was it a false positive (i.e. positively diagnosed by the health worker among those without malnutrition detected from anthropometric measurements of surveyors), or otherwise, was it a false negative (i.e. negatively diagnosed by the health workers among those suffering from acute malnutrition according to anthropometric measurements taken by surveyors)? Using clinical files and register transcripts, we analysed the number of MAM and SAM cases followed-up at the HC per semester. There were serious issues with MAM treatment supply (which was based on corn-soy blend mixed with sugar and oil) during the project and many HCs did not offer a MAM management service anymore; therefore, we also checked whether the PBF-N intervention had an impact on offering this service at all. As for growth promotion, we checked to what extent the growth curve was mentioned during consultations, and whether growth monitoring sessions were reported as regularly organised by the nutrition service managers in the HCs.

## Statistical analysis

Difference-in-difference estimates were performed in order to assess the impact of PBF-N on primary and secondary outcomes as well as on intermediary outcomes. The following model was used:

$$Y_{iht} = f(\beta.PBFnut_{ht} + \gamma.t + \delta.group_h + \varepsilon_{iht})$$

Where $i$ indicates the individual (when appropriate), $h$ the health centre, and $t$ the survey wave

(before or after the intervention); $Y_{iht}$ refers to the dependent variable (whether primary or secondary or intermediary outcome); $f(.)$ is the indicator function. The variable $PBFnut_{ht}$ is an interaction term multiplying group with period: it equals 1 only in the intervention HCs and after the intervention, and 0 otherwise; the beta coefficient provides an estimate of the impact of the PBF-N on the dependent variable. We controlled through dummy variables for time effects ($t$) that were common across all HCs, and group specific effects (intervention or control) that were common over time.

For binary variables, logit estimates were performed, and for the impact estimates, we reported odds ratios (OR) as well as marginal impacts when appropriate (i.e. the absolute change in a rate expressed in percentage points due to the intervention). For continuous variables, the Ordinary Least Square (OLS) estimator was used, except for the number of MAM and SAM cases managed at HC during the last semester where the Poisson estimator was used. For all estimates we used the sandwich estimator to take into account the clustering at HC level.

In displayed results, no control variable was added; however, as for robustness checks, equations using individual and HC (fixed or time varying) characteristics were included to gain precision ($X'_{iht}$), see model below:

$$Y_{iht} = f(\beta.PBFnut_{ht} + \sum (X'_{iht}.\theta) + \gamma.t + \delta.group_h + \varepsilon_{iht})$$

At HC level, control variables include dummies indicating (1) whether staff of the health centre was in line with national norms, (2) whether anthropometric measurements equipment was complete, (3) whether supervision was recently performed at the HC, (4) whether MAM treatment was available, (5) whether the MAM service was closed. For outcomes retrieved from the observed consultations or vignettes, HW related variables were used, such as their age, gender, salary level, whether salary was received, whether acute malnutrition management was in their terms of reference, whether they were supervised in the last six months. For outcomes retrieved from household data, we added a wealth quintile variable, age, gender, and education of household head, age and education of mother, age of the child, a food security index, a dietary index of the child related to the frequency and food groups, and the distance to HC.

Conditional effects were tested, using an interactive variable combining the $PBFnut_{ht}$ variable alternatively with control variables such as the time distance to the HC (less than 30 minutes versus more), whether health workers in the HC had been recently trained on nutrition, the quality of malnutrition screening in the HC (as found out from vignettes and from clinical files), wealth quintile of children, and the existence in the HC of a health worker dedicated to health promotion and in charge of supervising CHWs.

## Findings

In total, respectively 6,199 and 6,480 children aged 6–23 months were surveyed during the baseline and the endline household surveys. With the health facility baseline and endline surveys, we got respectively 971 and 674 MAM clinical files transcribed, and 963 and 1,043 SAM clinical files transcribed (out of 1,080 expected each time). The low rate of transcription for MAM clinical files within the endline survey is attributed to the control group, where only 29% of the expected files were found and transcribed, compared to 96% in the intervention group. This different compliance for the medical record can be interpreted as a first result of the PBF-N; it indeed stems from a different response in front of the discontinuation in the MAM treatment supply. Because of the PBF-N reward, HCs in the intervention group had still an incentive both to carry out MAM and to complete the clinical files (as data from the medical

files was used for the payment calculation); conversely, HCs in the control group had no incentive neither to look for strategies to circumvent the shortages nor to complete the patient files; in several control HCs, MAM services were even stopped for a while. We also collected information at the baseline and endline on respectively 515 and 529 observed paediatric curative consultations followed by exit interviews; as well as on the HC general and nutrition specific situation in the 90 HCs.

Baseline data found no significant difference between the two groups (intervention and control) as for the variables of primary interest for this research, whether at the HC or household levels. Among the 386 variables tested overall, we observed a significant difference between the intervention and control groups (p<0.05) for 26 (6.7%) of them (see details in the baseline report, S7 File). The findings showed that prevention and management of malnutrition were very problematic, technical capacities limited [25] and global malnutrition prevalence very high [26].

The PBF-N intervention started in January 2015, at the hospital and HC levels, and was delayed to October 2015 at the community level. The intervention suffered from implementation problems such as delays in payments to providers. A more important issue concerned the MAM treatment supply, which was discontinued at the time of the implementation; a local solution was initiated one year after the intervention started but suffered from communication and quality of product issues. Finally, political followed by economic instability affected the project. Nevertheless, the intervention occurred and HCs in the intervention group were paid accordingly to their performance in nutrition activities. More details on the implementation and on the context can be found in the endline report and in other papers [21, 25, 27].

## Findings at the health centre level

As explained above, we did not gather the 1,080 clinical files initially planned. Furthermore, in some patient files, the outcome as for the end of the treatment (recovered or not) was missing. At baseline, the MAM and SAM outcomes were reported in respectively 628 and 665 clinical files. This affected the explanatory power of our tests. At baseline, the ICC for MAM recovery outcome was assessed to be up to 0.41 (based on observations from 26 HC for which we had the outcome information for 12 clinical files), which reduced the explanatory power to 60% for a recovery rate of increase from 80% to 95%. The ICC for SAM was estimated at 0.06, and the explanatory power remained at 80% with an $\alpha$-error of 5% assuming that the intervention would result in increasing the recovery rate of SAM from 85% to 95%.

Management of cases has improved thanks to the intervention. MAM recovery rate increased from 84% in 2014 to 97% in 2017 in the intervention group, whereas it stayed the same in the control group (respectively 76% and 78%). The logit estimates display a significant positive impact of the PBF-N of 5.59 in terms of odds ratio (p = 0.039) and +14.7 percentage points in terms of marginal impact on the recovery rate (p = 0.007). This may be underestimated since only a few clinical files have been collected in the control group. We indeed think that the bias is towards underestimates, as the HCs in the control group that kept their MAM management services running were probably the most committed to the nutrition domain and/or those assisted by aid actors (most of MAM clinical files in the control group come from two provinces assisted by the World Food Program). The treatment duration for the recovered patients also significantly improved according to the OLS estimates (-33 days, p = 0.040). Uncomplicated SAM recovery rate increased from 84% to 92% in the intervention group and from 87% to 93% in the control group, and impact was not found significant (OR: 1.16, p = 0.751). However, uncomplicated SAM treatment duration among the recovered patients also significantly improved thanks to the intervention (-20 days, p = 0.025; cf. Table 2).

**Table 2. Impact of PBF-N on MAM and uncomplicated SAM services' performance.**

| | | Control group | | | Intervention group | | | PBF-N impact | | | |
|---|---|---|---|---|---|---|---|---|---|---|---|
| | | | | | | | | Logit estimates | | | |
| | | % | n | N | % | n | N | Odds ratio | p-value | (95% CI) | Nb of obs |
| **MAM recovery rate** | before | 76% | 234 | 308 | 84% | 268 | 320 | 5,59 | 0,039 | (+1,09; +28,70) | 1 067 |
| | after | 78% | 87 | 112 | 97% | 317 | 327 | | | | |
| **Uncomplicated SAM recovery rate** | before | 87% | 306 | 352 | 84% | 262 | 313 | 1,16 | 0,751 | (0,46; 2,92) | 1 402 |
| | after | 93% | 359 | 387 | 92% | 322 | 350 | | | | |
| | | | | | | | | OLS estimates | | | |
| | | Mean | SD | N | Mean | SD | N | Coef. | p-value | (95% CI) | Nb of obs |
| **Treatment duration among the MAM cases who recovered** | before | 70.79 | 47.18 | 225 | 78.08 | 53.78 | 252 | -33.59 | 0.040 | (-65.5; -1.7) | 842 |
| | after | 70.26 | 39.82 | 77 | 43.96 | 40.55 | 288 | | | | |
| **Treatment duration among the uncomplicated SAM cases who recovered** | before | 56.97 | 41.22 | 304 | 61.19 | 34.16 | 252 | -20.42 | 0.025 | (-38.3; -2.5) | 1 155 |
| | after | 59.06 | 41.08 | 317 | 42.87 | 32.35 | 282 | | | | |

These estimates correspond to model (1); a model using control variables were also tested, leading to a higher explanatory power and similar coefficients and p-values regarding the PBF-N impact. Meanings: % is proportion, n is number of occurrences, SD is standard deviation, N is number of observations, and CI is confidence interval.

Similarly, PBF-N had a significant impact on the number of children treated for acute malnutrition. According to our Poisson estimates, the intervention had a multiplying effect of 17.9 on the number of MAM cases per semester per health centre (p<0.001), and of 3.06 on the number of SAM cases per semester per health centre (p<0.001). Yet the intervention did not have any influence on malnutrition screening during paediatric curative consultations. In 2017, only 18 cases were diagnosed as suffering from malnutrition in the regular consultation, while according to the anthropometric data collected during the exit interview, 70 children were malnourished (out of 528 cases). We assessed the impact of the intervention on false positive cases of acute malnutrition (AM) among the non-AM cases according to the anthropometrics taken at exit interview, and on false negative cases among the AM cases: no significant impact was found on false positive and false negative rates. The 2017 fictive vignette cases confirmed the appalling situation with capacities of health workers in AM diagnosis already observed in 2014 [25]: only three HWs in 2014 and two in 2017 succeeded in diagnosing the three acute malnutrition cases correctly (out of 145 in 2014 and 168 in 2017). Unsurprisingly we found no significant impact of the intervention on diagnosis success rate, whatever the vignette (cf. Table 3). Findings do not show either any influence of the PBF-N intervention on activities on growth monitoring, whatever the source of information: the growth curve is mentioned in 4% of consultations in 2017 (even less than in 2014), it is drawn in less than 3% of health booklets of children surveyed in the household survey (three times less than in 2014; cf. Table 4).

## Findings at the community level

The improvements of malnutrition management at the HC level did not translate into better results at the community level. Prevalence of chronic malnutrition among children aged 6–23 months remained high, above 50%. Prevalence of acute malnutrition increased from 6% in December 2014 (6.3% in control group, 5.8% in intervention group) to 8.8% in March 2017

**Table 3. Impact of PBF-N on acute malnutrition diagnosis and screening.**

| | | Control group | | | Intervention group | | | PBF-N Impact | | | |
|---|---|---|---|---|---|---|---|---|---|---|---|
| | | | | | | | | Logit estimates | | | |
| | | % | n | N | % | n | N | Odds ratio | p-value | (95% CI) | Nb of obs |
| **Within consultations observed:** | | | | | | | | | | | |
| **Acute malnutrition was diagnosed** | before | 3.46% | 9 | 260 | 1.97% | 5 | 254 | 1.77 | 0,527 | (0,30; 10,42) | 1 042 |
| | after | 3.42% | 9 | 9263 | 3.40% | 9 | 265 | | | | |
| **False positive of acute malnutrition diagnosis** | before | 1.00% | 2 | 205 | 1.33% | 3 | 226 | 2.98 | 0,512 | (0,11; 78,34) | 876 |
| | after | 0.45% | 1 | 223 | 1.80% | 4 | 260 | | | | |
| **False negative of acute malnutrition diagnosis** | before | 86.54% | 45 | 52 | 92.59% | 25 | 27 | 0.98 | 0,983 | (0,11; 8,50) | 162 |
| | after | 80.00% | 32 | 40 | 88.37% | 38 | 43 | | | | |
| **Within fictive cases:** | | | | | | | | | | | |
| **Vignette 1: diagnosis of complicated SAM was found** | before | 7.0% | 5 | 71 | 6.8% | 5 | 74 | 1,34 | 0,718 | ('0,27; 6,67) | 313 |
| | after | 12.9% | 11 | 85 | 16.0% | 13 | 81 | | | | |
| **Vignette 2: diagnosis of uncomplicated SAM was found** | before | 15.5% | 11 | 71 | 12.2% | 9 | 74 | 1,85 | 0,384 | (0,46; 7,40) | 313 |
| | after | 20.0% | 17 | 85 | 25.9% | 21 | 81 | | | | |
| **Vignette 3: diagnosis of MAM was found** | before | 28.2% | 20 | 71 | 24.3% | 18 | 74 | 0,83 | 0,722 | (0,30; 2,29) | 313 |
| | after | 42.4% | 36 | 85 | 33.3% | 27 | 81 | | | | |
| **Using clinical files transcripts:** | | | | | | | | Poisson estimates | | | |
| | | Mean (median) | SD | N | Mean (median) | SD | N | IRR | p-value | (95% CI) | Nb of obs |
| **Nb of MAM cases per semester** | before | 62.0 (29) | 109.4 | 45 | 33.4 (20) | 39.5 | 45 | 17,9 | <0.001 | (7,11; 45,07) | 180 |
| | after | 12.7 (0) | 27.1 | 45 | 122.2 (66) | 182.7 | 45 | | | | |
| **Nb of uncomplicated SAM cases per semester** | before | 40.1 (33) | 36.4 | 45 | 24.9 (17) | 21.4 | 45 | 3,06 | <0.001 | (1,92; 4,87) | 180 |
| | after | 41.5 (28) | 46.4 | 45 | 78.9 (62) | 68.8 | 45 | | | | |

These estimates correspond to model (1); a model using control variables were also tested, leading to a higher explanatory power and similar coefficients and p-values regarding the PBF-N impact. Meanings: % is proportion, n is number of occurrences, SD is standard deviation, N is number of observations, IRR is incidence rate ratio and CI is confidence interval.

**Table 4. Impact of PBF-N on growth monitoring activities.**

| | | Control group | | | Intervention group | | | PBF-N impact | | | |
|---|---|---|---|---|---|---|---|---|---|---|---|
| | | | | | | | | Logit estimates | | | |
| | | % | n | N | % | n | N | Odds ratio | p-value | (95% CI) | Nb of obs |
| **Growth curve mentioned in consultations observed** | before | 6.2% | 16 | 260 | 8.3% | 21 | 254 | 0.80 | 0.776 | (0,17; 3,80) | 1 042 |
| | after | 3.8% | 10 | 263 | 4.2% | 11 | 265 | | | | |
| **Growth monitoring sessions are reported as regularly organized** | before | 80.0% | 36 | 45 | 84.4% | 38 | 45 | 2,47 | 0,200 | (0,62; 9,82) | 180 |
| | after | 51.1% | 23 | 45 | 77.8% | 35 | 45 | | | | |

These estimates correspond to model (1); a model using control variables were also tested, leading to a higher explanatory power and similar coefficients and p-values regarding the PBF-N impact. Meanings: % is proportion, n is number of occurrences, N is number of observations and CI is confidence interval.

**Table 5. Impact of PBF-N at the community level.**

| | | Control group | | | Intervention group | | | PBF-N impact | | | |
|---|---|---|---|---|---|---|---|---|---|---|---|
| | | | | | | | | Logit estimates | | | |
| | | % | n | N | % | n | N | Odds ratios | p-value | (95% CI) | Nb of obs |
| **Acute malnutrition prevalence** | before | 6.22% | 193 | 3100 | 5.78% | 179 | 3099 | 1,08 | 0,628 | (0,79; 1,47) | 12 679 |
| | after | 8.73% | 283 | 3240 | 8.70% | 282 | 3240 | | | | |
| **Chronic malnutrition prevalence** | before | 52.97% | 1642 | 3100 | 53.21% | 1649 | 3099 | 1,01 | 0,893 | (0,88; 1,16) | 12 679 |
| | after | 49.88% | 1616 | 3240 | 51.91% | 1682 | 3240 | | | | |
| | | | | | | | | OLS estimates | | | |
| | | Mean | SD | N | Mean | SD | N | Coef | p-value | (95% CI) | Nb of obs |
| **Weight for length z-score** | before | -0.34 | 1.09 | 3100 | -0.34 | 1.08 | 3098 | +0.02 | 0.725 | (-0.08;+0.12) | 12 677 |
| | after | -0.47 | 1.12 | 3233 | -0.46 | 1.14 | 3246 | | | | |
| **Length for age z-score** | before | -2.10 | 1.22 | 3100 | -2.11 | 1.24 | 3099 | -0.01 | 0.783 | (-0.11;+0.08) | 12 673 |
| | after | -2.06 | 1.30 | 3230 | -2.08 | 1.23 | 3240 | | | | |
| **Mid-Upper Arm Circumference (MUAC)** | before | 139.75 | 12.42 | 3100 | 140.10 | 12.32 | 3099 | -0.07 | 0.893 | (-1.1;+1.0) | 12 678 |
| | after | 137.96 | 12.36 | 3239 | 138.24 | 12.66 | 3240 | | | | |
| | | | | | | | | Logit estimates | | | |
| | | % | n | N | % | n | N | Odds ratios | p-value | (95% CI) | Nb of obs |
| **Among AM cases, follow-up at the HC for AM** | before | 8.81% | 17 | 193 | 8.38% | 15 | 179 | 1,53 | 0,341 | (0,64; 3,67) | 937 |
| | after | 13.78% | 39 | 283 | 18.79% | 53 | 282 | | | | |
| **Among the SAM cases, follow-up at the HC for AM** | before | 12.90% | 4 | 31 | 19.51% | 8 | 41 | 1,08 | 0,935 | (0,18; 6,43) | 191 |
| | after | 15.00% | 9 | 60 | 23.73% | 14 | 59 | | | | |

These estimates correspond to model (1); a model using control variables were also tested, leading to a higher explanatory power and similar coefficients and p-values regarding the PBF-N impact. Meanings: % is proportion, n is number of occurrences, SD is standard deviation, N is number of observations, and CI is confidence interval.

(8.8% in control group, 8.7% in intervention group); we attribute this increase to seasonality and, possibly, to the general deterioration of the country's economy. PBF-N had any significant impact on neither chronic nor acute prevalence rates (cf. Table 5).

We also checked with the household survey data whether more malnourished children were under HC management. It appeared that, among the children identified by our surveyors with acute malnutrition, only a few of cases were managed in HCs: less than 10% in both groups in 2014, and 13.8% and 18.8% respectively in the control and intervention groups in 2017. We found a non-significant impact of the PBF-N intervention of 1.53 OR (p = 0.341) on the management rate (cf. Table 5). Using a set of control variables improved the explanatory power of the models but the PBF-N impact coefficient remained non significantly different from zero (cf. Table 6).

## Discussion

### Summary of findings

This study is to our knowledge, the first cRCT assessing the effect of an extension of a PBF scheme to nutrition services. Our findings indicate that the PBF-N scheme piloted over 2015–2016 in rural Burundi improved malnutrition management at HC level, despite serious resource and capacity constraints. Thanks to the PBF-N incentives, there was an improvement in the outcome of the treatment: MAM recovery rates increased by 14.7 percentage points (p = 0.007), while MAM and uncomplicated SAM duration of treatment was reduced by respectively 34 and 20 days (resp. p = 0.040 and p = 0.025), suggesting more efficient practices.

**Table 6. Conditional impact of PBF-N on follow-up at the health centre for acute malnutrition.**

| Logit estimates | | Follow-up at the HC for AM (yes/no) | | | | | |
|---|---|---|---|---|---|---|---|
| | | Among both MAM and SAM cases | | | Among SAM cases only | | |
| PBF-N impact | Odds ratio | 1,24 | 0,97 | 0,7 | 1,26 | 1,06 | 0,56 |
| | p-value | 0,685 | 0,957 | 0,525 | 0,843 | 0,964 | 0,655 |
| PBF-N impact x Short distance | Odds ratio | | 2,47 | | | 1,9 | |
| | p-value | | 0,058 | | | 0,544 | |
| PBF-N impact x supervision of | Odds ratio | | | 3,9 | | | 34,31 |
| CHWs at HC level | p-value | | | 0,001 | | | 0,003 |
| Control variables | | Yes | Yes | Yes | Yes | Yes | Yes |
| Nb of observations | | 712 | 712 | 712 | 141 | 141 | 141 |

These logit estimates correspond to model (1), using in addition control variables and interactive variables.

There was also an increase in the number of malnourished children managed by the HCs (see in the evaluation report, S8 File). Yet, the volume increase at facility level was not massive enough to generate large benefits at population level: we did not find any significant impact on acute and chronic malnutrition prevalence among children aged 6–23 months. This is in line with other studies on PBF in Burundi, which also reported improvements in service utilisation [10, 12, 28] and quality of care [11, 29], but not always health benefits at population level [10, 29, 30].

Further data analyses found out that there were several missed opportunities, a major one —the insufficient effort to detect children during the general paediatric consultation—directly stems from a lack of knowledge and know-how at the level of health workers [31]. This weak capacity, already present at the baseline, has not been modified by the intervention. In addition, growth monitoring remains a marginal activity while it is in fact essential to growth promotion, malnutrition screening, and, *in fine*, to the reduction of child malnutrition [32]. Further analyses displayed in the final report detect that the PBF-N intervention was not enough to tackle several other structural constraints (e.g. sets of anthropometrics' equipment -with scales, measuring boards, WHZ tables, etc.- were not fully available and functional in most of the HCs), at least not within the two-year period of the study. The discontinuation of the MAM treatment supply as well as the political and financial instability also hindered the PBF-N to operate [21, 24, and S8 File].

At community level, prevalence of chronic malnutrition among children aged 6–23 months remains high, above 50%, and there was no impact of the intervention on neither chronic nor acute malnutrition prevalence. This is not surprising, as malnutrition is multifactorial, and prevention, identification and management of malnutrition at health system level can compensate but cannot fix food security nor water, sanitation and hygiene failures. Burundi is highly dependent on agriculture [33], 91% of our surveyed households suffered from moderate to severe food insecurity, and about three quarters of children aged 6–23 months did not have a diet that met the required diversity and frequency. Improved water source and sanitation are accessible by respectively 82% and 39% of rural inhabitants [4] and according to our findings, 65% of them had soap. PBF-N did not address these issues, mainly because this goes beyond its scope: this is not a mechanism that can address socio-economic determinants such as wealth, education of mothers and number of children [26].

Still, among the children identified with acute malnutrition by our surveyors, too few were managed by the HCs. According to us, the main priority should be to increase the number of detected children: at community level, among children aged 6–23 months, there were still, at

the endline, more than 80% of AM cases not followed-up in HCs in the intervention group. The improvement in screening should happen both at community and facility levels. At facility level, during paediatric consultations, health workers' practices towards malnutrition detection were found to be weak, and it appeared that PBF-N had no impact on this: the first constraint probably is health workers' technical capacities. Nevertheless, our study indicates a strategy which seems particularly powerful: PBF-N had a significant impact on the volume of detected children in HCs where there was a health worker dedicated to health promotion and in charge of supervising CHWs. This indicates that community health could be a central strategy for the screening, detection and referral of acute malnutrition cases. This finding also suggests that the CHW strategy did get a boost from the PBF-N intervention, both at community and HC level, i.e. both through the provision of revenues, conditional on performance, to CHWs, and the PBF-N intervention at the health centre level which encouraged some HCs to work with and integrate CHWs to their malnutrition screening activity. This kind of supervision of CHWs should be scaled up.

This study was based on a pilot project, intended to inform policy decisions related to health financing and nutrition. To our mind, the main policy lessons do not directly relate to PBF-N: they bear on fundamental weaknesses of the health system. There is a need for much more attention to malnutrition management in Burundi. Many children could benefit from MAM or SAM services. The priority is to detect those children and deliver them quality services. Better detection will come from improved collaboration with and supervision of CHWs, growth monitoring and capacity to perform integrated management of childhood illnesses, with having personnel specifically in charge of health promotion. Staff technical capacities are a bottleneck, and there is an urgent need for specific training and closer supervision. Issues with supplement shortages are another major constraint and a source of staff demotivation (what is the point to screen malnourished children if you have nothing to give them?).

Our findings have also relevance beyond Burundi. There is a case for expanding PBF schemes to nutrition services in some other low-income countries, but there must be a clear understanding that PBF will not suddenly solve all systemic weaknesses: in many poor-resource settings, complementary health system strengthening efforts will probably be needed. There is also a need for more research to confirm our findings and their validity in other contexts.

## Limitations

This impact evaluation followed a *cRCT design*–this ensures robust findings. However, results coming from the MAM management clinical files should be taken with some caution because of the attrition in the control HCs. Because of the lack of food treatment, most control HCs have closed the MAM service; the only ones that kept the service running were those helped by other aid organisations or for which nutrition was a priority (for instance, because of some intrinsic motivation at the level of the staff). This selection bias among control HCs suggests that the effect of PBF-N on treatment outcome is under-estimated. At the community level, the short duration of the intervention (two years) may not have allowed to show an impact on chronic malnutrition, although we believe other factors than the health system itself, like food security, water, sanitation and hygiene elements, are much more structural determinants.

## Conclusion

Child malnutrition is a major health issue in Burundi. It also compromises the present and future economic development of the country. Major action is required. Our findings indicate that PBF can be part of the strategy. Yet, its effectiveness will gain from a general improvement

of malnutrition screening and identification, both at health centre and community levels. Built-in in a health system approach, with improved coordination with CHWs, capacities of HWs and nutrition services, its effects should be more important than what we documented in this pilot study. Still, PBF won't be enough. Malnutrition in Burundi calls for much broader and multisectoral effort from a large national and international coalition.

## Supporting information

**S1 Table. Baseline characteristics of households and children.** Notes: N refers to the number of observations; n refers to the number of positive (yes to a categorical variable); SD refers to the standard deviation. Source: Authors.
(DOCX)

**S1 Fig. Chain of results with PBF-N.** Source: Authors.
(TIF)

**S1 File. CONSORT checklist.** Source: Authors.
(PDF)

**S2 File. Quality indicators specific to PBF-N.** Source: MSPLS. Note technique relative à l'intégration de la nutrition dans la stratégie nationale de financement basé sur la performance. 2013.
(PDF)

**S3 File. Study protocol, household level, as approved by the University of Antwerp Ethics Committee, in French (original language).** Source: Authors.
(PDF)

**S4 File. Study protocol summary in English, household level, as approved by the University of Antwerp Ethics Committee.** Source: Authors.
(PDF)

**S5 File. Study protocol, health centre level, as approved by the University of Antwerp Ethics Committee, in French (original language).** Source: Authors.
(PDF)

**S6 File. Study protocol summary in English, health centre level, as approved by the University of Antwerp Ethics Committee.** Source: Authors.
(PDF)

**S7 File. Baseline report, in French: « FBP Nutrition au Burundi.** Rapport des enquêtes de référence au niveau des centres de santé et des ménages ». Source: Authors.
(PDF)

**S8 File. Impact evaluation report, in French: « FBP Nutrition au Burundi.** Evaluation d'impact au niveau des centres de santé et des ménages ». Source: Authors.
(PDF)

**S1 Dataset. Synthesis of datasets used for analysis, with summary statistics (mean, median, sum, variance, standard deviation, number of observations) by survey and treatment group.** Source: Authors.
(XLSX)

## Acknowledgments

We thank all MoH members (especially Jean Kamana and Olivier Basenya and their colleagues from the PBF committee and Nutrition program) for their key role in implementing the intervention and their collaboration for the study, Driss Zine-Eddine El-Idrissi and Alain-Désiré Karibwami from the World Bank for their facilitation, INSP and ISTEEBU teams for implementing the surveys with success, Epco Hasker and Tom Smekens for their statistics advice. The implementation of this study also benefited from the assistance of our colleagues Kirrily de Polnay, Maxime Rouve, Ulises Huerta, Elodie Macouillard, Léonard Ntakarutimana and Jacqueline Manisabwe.

## Author Contributions

**Conceptualization:** Catherine Korachais, Bruno Meessen.

**Data curation:** Catherine Korachais.

**Formal analysis:** Catherine Korachais, Manassé Nimpagaritse.

**Funding acquisition:** Catherine Korachais.

**Investigation:** Catherine Korachais, Sandra Nkurunziza, Manassé Nimpagaritse.

**Methodology:** Catherine Korachais, Bruno Meessen.

**Project administration:** Catherine Korachais, Manassé Nimpagaritse.

**Resources:** Catherine Korachais.

**Supervision:** Catherine Korachais, Bruno Meessen.

**Validation:** Catherine Korachais, Sandra Nkurunziza, Manassé Nimpagaritse, Bruno Meessen.

**Visualization:** Catherine Korachais.

**Writing – original draft:** Catherine Korachais, Sandra Nkurunziza, Manassé Nimpagaritse, Bruno Meessen.

**Writing – review & editing:** Catherine Korachais, Sandra Nkurunziza, Manassé Nimpagaritse, Bruno Meessen.

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
