## [Decision Letter · Decision Letter 0]

7 Apr 2020

PONE-D-19-30259

Impact of the extension of a performance-based financing scheme to nutrition services in Burundi on malnutrition prevention and management among children below five: a cluster-randomized control trial

PLOS ONE

Dear Dr. Korachais,

Thank you for submitting your manuscript to PLOS ONE. After careful consideration, we feel that it has merit but does not fully meet PLOS ONE’s publication criteria as it currently stands. Therefore, we invite you to submit a revised version of the manuscript that addresses the points raised during the review process.

We would appreciate receiving your revised manuscript by May 22 2020 11:59PM. To enhance the reproducibility of your results, we recommend that if applicable you deposit your laboratory protocols in protocols.io, where a protocol can be assigned its own identifier (DOI) such that it can be cited independently in the future. For instructions see: http://journals.plos.org/plosone/s/submission-guidelines#loc-laboratory-protocols

We look forward to receiving your revised manuscript.

Kind regards,

Seth Adu-Afarwuah

Academic Editor

PLOS ONE

Journal Requirements:

2. In your Methods section, please provide additional information about the participant recruitment method and the demographic details of your participants. Please ensure you have provided sufficient details to replicate the analyses such as: a) the recruitment date range (month and year), b) a description of any inclusion/exclusion criteria that were applied to participant recruitment, c) a table of relevant demographic details, d) a statement as to whether your sample can be considered representative of a larger population, e) a description of how participants were recruited, and f) descriptions of where participants were recruited and where the research took place.

3. Please correct your reference to "p=0.000" to "p<0.001" or as similarly appropriate, as p values cannot equal zero.

4. Please provide additional details regarding participant consent. In the ethics statement in the Methods and online submission information, please ensure that you have specified (1) whether consent was informed and (2) what type you obtained (for instance, written or verbal). If your study included minors, state whether you obtained consent from parents or guardians. If the need for consent was waived by the ethics committee, please include this information.

"CK, SN and MN declare no conflict of interest.

BM has contributed to the emergence of PBF as a global health policy, through technical assistance, research and knowledge management. He is the lead facilitator of the PBF Community of Practice. He holds minority shares in Blue Square, a firm developing health system software solutions. "

6. Please remove your figures from within your manuscript file, leaving only the individual TIFF/EPS image files, uploaded separately.  These will be automatically included in the reviewers’ PDF.

Reviewers' comments:

Reviewer's Responses to Questions

**Comments to the Author**

1. Is the manuscript technically sound, and do the data support the conclusions?

Reviewer #1: Partly

Reviewer #2: Yes

Reviewer #3: Yes

2. Has the statistical analysis been performed appropriately and rigorously? 

Reviewer #1: No

Reviewer #2: Yes

Reviewer #3: I Don't Know

3. Have the authors made all data underlying the findings in their manuscript fully available?

Reviewer #1: No

Reviewer #2: Yes

Reviewer #3: No

4. Is the manuscript presented in an intelligible fashion and written in standard English?

Reviewer #1: Yes

Reviewer #2: Yes

Reviewer #3: Yes

5. Review Comments to the Author

Reviewer #1: This paper reported an important national study that aimed to address two research questions: (1) did the extension of performance-based financing to nutrition (PBF-N) scheme contributed to improve nutritional health services for children below five years old at HC level and; (2) did it reduce the prevalence of malnutrition among children below two years old at community level? Of the 193 eligible health centres providing nutrition services in Burundi, 100 HCs were randomised and 90 were selected to the study. Cross-sectional household surveys were conducted separately at baseline and at end-line about two years after the PBF-N intervention has started. In addition, health centre surveys were performed at both visits and routine data were obtained from the PBF-N program. The primary outcomes were the prevalence of chronic and acute malnutrition among children aged 6-23 months measured at the community level, and the MAM/SAM recovery rates among children below five years old at the HC level.

The study has followed a cluster-randomised controlled trial design, with half of the 90 selected HCs randomly allocated to intervention and the other half constituted the control group. Data were collected at both cluster and individual levels using two cross-sectional surveys. The methods were well documented; however, I have major concerns on the overall trial design, statistical analysis and results presented in this paper.

The CONSORT flow chart suggested that 193 HCs were assessed for eligibility, 100 were randomised, and 90 were selected and enrolled to the study. The authors did not explain why only half of the HCs were randomised, and the discrepancy between 100 and 90 HCs that were randomised. In principle, only eligible clusters should be randomised and included in final analysis.

The sample size of the household surveys was based on a reduction of 25% in acute malnutrition prevalence between HCs in the two groups. The number of children recruited in each HC was set to 72, from 12 sub-hills per cluster and 6 eligible households per sub-hill. This adds up to a total of 6,480 children aged 6-23 months surveyed at the community level. Was the same target used for baseline survey, which as reported included a total of 6,199 children?

For a cluster randomised trial, sample size calculation should be based on both the number of clusters and average cluster size with pre-specified statistical power (>80%) and significance level (normally 5%). Another important parameter to consider is the inflated design effect related to the intra-cluster correlation coefficient (ICC), which is the proportion of the total variance of the outcome that can be explained by the variation between clusters. The flow chart should also include information on both clusters and individuals, as recommended by the CONSORT 2010 statement with extension to cluster randomised trials.

For outcome analysis, the statistical model has included both time points while only the end-line survey data were potentially impacted by the intervention. The beta coefficients estimated the difference-in-difference, which would be hard to quantify the actual effect sizes between intervention and control groups on the defined outcomes. A standard approach is to analyse the end-line survey data only using the mixed models with a random cluster effect, with the assumption that the clusters were similar at baseline by randomisation and matching.

As required by the CONSORT guidelines, baseline characteristics at cluster and individual participant levels should be presented in a table for each group. Continuous variables can be summarised by the mean and standard deviation, and numbers and proportions should be reported for categorical variables. The supplementary table 1 needs to be modified following the guidelines. The outcome tables are hard to follow, and the numbers are inconsistent. For example, how were the total numbers of observation 1,254 and 1,549 obtained in Table 2? The column titles Mean and N were used for all outcomes regardless of their distributions. With an interaction term between group and period in the main model, which term was used to estimate the effects of intervention on different outcomes should be clearly stated so that the estimated coefficients, odds ratios, and incidence rate ratios can be interpreted appropriately. With more than one primary outcome and a large number of variables tested overall (386), any significant differences observed between intervention and control groups should be concluded with caution without controlling for overall Type 1 error rate.

Reviewer #2: 1. Abstract. The word “data” is plural so it should be “data were”.

2. p. 9. The last paragraph purports to explain the theory of change of the intervention. It does not do so. Having this explanation is essential especially for understanding why the authors expected health system improvements to have reduced chronic or acute malnutrition prevalence, an expectation that on the face of it does not seem warranted because health systems cannot remove most of the causes of the malnutrition (as the authors note in the Discussion). What the authors have provided here is useless for explaining the theory of change. The authors have basically said to the reader to go find out for yourself by going to the supplementary material, which is irritating and not acceptable. S2-S5 appear to have two files duplicated twice. The protocol is in French; PLoS One is an English language publication. Please revise to explain clearly in the main text of the manuscript the theory of change of the intervention for both recovery from and prevention of malnutrition.

3. p. 14, top. The authors write “We talk about shortness when LAZ is lower than minus two standard deviations (SD) and stunting when it’s lower than minus three SDs.” WHO (and just about everyone else) defines stunting as less than -2 SD, not -3 SD. Please revise and use the standard definitions to avoid confusion.

4. p.15. Where it says “The time varying variable PBFnutht is an an interaction term…” delete “time varying”. This term is not time varying in the common use of this descriptor.

5. p. 22. P-values, by definition, cannot be zero. All p-values given as 0.000 should be given as <0.001.

6. The manuscript requires careful technical editing and preparation. Overall, the writing is understandable, but many technical problems are throughout the manuscript. Furthermore, the manuscript was not carefully prepared, for example, line numbers in only part of the manuscript.

Reviewer #3: • General comments

This is a quality paper on an important nutrition issue using state-of-the art methodology (RCT) with particularly careful pairing of control and intervention HCs. It has often been assumed that health personnel and workers did not have enough incentives to devote more effort to nutrition services and therefore, it was hypothesized that performance-based monetary incentives would boost the nutrition performance and impact. There are some issues, however minor, that need to be addressed.

• Specific comments

1. Abstract

The objectives should be given. The results section is not very clear : 97% - 78% = 19%, not 14.7pp (what is “pp”?). Unless this is not the way the results should read.

2. Introduction :

a. Hunger and food insecurity are among the root causes of malnutrition. Malnutrition encompasses several forms including undernutrition. In addition to poverty and food insecurity, there are factors of child undernutrition which are in the health realm, beyond nutrition care and services, and management of undernutrition, that is, poor access to health care as well as WASH. This could be clearer in the paper.

b. Community-base management of undernutrition (and prevention) is now advocated, although it is clear that health personnel are involved, at community, as well as HC and hospital level. The expected nutrition activities and services at these three levels should be described, perhaps in general and in the Burundi context.

3. Methods

a. How many health centers in total for 200 providing nutrition services?

b. The rationale for the different age range in the health center and the community assessment should be given.

c. Fig. 1: The unit is the health center, please specify

d. Institut national de santé publiQUE

e. Who gets the subsidy and how is it shared in the case of hospitals, HCs and CHWs?

f. What is the MAM treatment supply? It should be explicit as it is often referred to.

g. Shortness and stunting: are these widely accepted cut-offs and terms? Above the age of 24 months, it should be stated that standing height is measured, not length.

h. Please clarify that the duration of the intervention was two years.

i. Is it possible to provide the list of the 19 indicators of quality of nutrition services, perhaps as supplementary data?

j. According to Table 2, CHWs are to screen and refer acute malnutrition cases to HCs. However, is this not accomplished through growth monitoring and promotion activities, which are HC’s activities?

k. The protocol is not needed even in appendix; besides, it is in French.

4. Results

a. On p. 18, there is a distinction of the intervention at hospital, HC, and community level. This is why the specific activities expected at these three levels have to be described in the methods section.

b. Fig. 3 is not very useful.

c. What about the % of defaulters in MAM management, in intervention vs control HCs?

d. It has to be clear that the improved SAM management performance only refers to uncomplicated SAM treatment outside the hospital since control hospitals also get PBF for the management of SAM cases with complications.

e. In Table 2, what is the meaning of **?

f. What about infant and child feeding practices in the community survey? It is referred to it in the discussion (p. 32, line 29) but the results are not shown.

5. Discussion and conclusion

a. Training of health personnel (although recent training apparently had no impact) and supervision of CHWs would appear to be major conditions of performance that the PBF strategy did not address. This should be discussed in more depth.

b. In the recommendations, the need for health centre staff training and CHW supervision should be emphasized, along with having personnel specifically for health promotion.

c. Is the fact the control hospitals and HCs also received subsidies not a limitation of the study, although this was intended to clearly identify the impact of the monetary incentives’?

d. The short duration of the intervention (2 years) may not have allowed to show an impact on stunting; this can be regarded as a limitation of the study.

e. P. 31, line 19: reference needed

f. P. 31, line 21: What equipment?

g. P. 32, line 27: also failures in WASH, not only food security

h. P. 32, line 40: “CHW strategy got a boost from PBF”: Please explain

6. PLOS authors have the option to publish the peer review history of their article (what does this mean?). If published, this will include your full peer review and any attached files.

Reviewer #1: No

Reviewer #2: No

Reviewer #3: No

---

## [Author Response · Author response to Decision Letter 0]

18 Jun 2020

Dear editor and reviewers, 

Thank you for your feedbacks, we appreciated the detailed and committed review. We considered and took care of each comment, and feel that this made the paper clearer. You can appreciate our responses in the "responses to reviewers" document (18 pages), and modifications of the text in the revised manuscript. 

Sincerely,

the authors

---

## [Decision Letter · Decision Letter 1]

28 Jul 2020

PONE-D-19-30259R1

Impact of the extension of a performance-based financing scheme to nutrition services in Burundi on malnutrition prevention and management among children below five: a cluster-randomized control trial

PLOS ONE

Dear Dr. Korachais,

Thank you for submitting your manuscript to PLOS ONE. After careful consideration, we feel that it has merit but does not fully meet PLOS ONE’s publication criteria as it currently stands. Therefore, we invite you to submit a revised version of the manuscript that addresses the points raised during the review process.

We look forward to receiving your revised manuscript.

Kind regards,

Seth Adu-Afarwuah

Academic Editor

PLOS ONE

Reviewers' comments:

Reviewer's Responses to Questions

**Comments to the Author**

1. If the authors have adequately addressed your comments raised in a previous round of review and you feel that this manuscript is now acceptable for publication, you may indicate that here to bypass the “Comments to the Author” section, enter your conflict of interest statement in the “Confidential to Editor” section, and submit your "Accept" recommendation.

Reviewer #1: All comments have been addressed

Reviewer #3: All comments have been addressed

2. Is the manuscript technically sound, and do the data support the conclusions?

Reviewer #1: (No Response)

Reviewer #3: Yes

3. Has the statistical analysis been performed appropriately and rigorously? 

Reviewer #1: (No Response)

Reviewer #3: Yes

4. Have the authors made all data underlying the findings in their manuscript fully available?

Reviewer #1: (No Response)

Reviewer #3: Yes

5. Is the manuscript presented in an intelligible fashion and written in standard English?

Reviewer #1: (No Response)

Reviewer #3: Yes

6. Review Comments to the Author

Reviewer #1: (No Response)

Reviewer #3: Only two minor points:

1. In the abstract, HC should be defined. Perhaps the conclusion should be clearer that performance-based financing had not substantive effect on the prevention of malnutrition and therefore alternative or additional strategies are required.

2. If the protocol is already published [17], one wonders why it is provided in supplementary files (S3-S6).

7. PLOS authors have the option to publish the peer review history of their article (what does this mean?). If published, this will include your full peer review and any attached files.

Reviewer #1: No

Reviewer #3: No

---

## [Author Response · Author response to Decision Letter 1]

28 Jul 2020

The two reviewers who responded reported that all comments had been addressed. We are happy that we have responded to all your questions and requests and that we got to a common understanding on this work. 

There were two additional minor comments sent by reviewer #3. Here below are the two minor comments and our answers to them:

1 / His first comment was “In the abstract, HC should be defined. Perhaps the conclusion should be clearer that performance-based financing had not substantive effect on the prevention of malnutrition and therefore alternative or additional strategies are required.” We would like to thank him, that was indeed a problem. We replaced “HC” by “health centres (HC)” at its first appearance in the abstract. We also added a conclusive sentence to the abstract: “PBF can contribute to a better management of malnutrition at HC level; yet, to address the huge problem of child malnutrition in Burundi, additional strategies are urgently required.”

2/ His second comment was: “If the protocol is already published [17], one wonders why it is provided in supplementary files (S3-S6).” We added these supplementary files because this corresponds to the protocol that was approved by the ethics committee, and this is a requirement from PLOS ONE to provide such a copy. These files are not accessible from the published version of the protocol.

---

## [Decision Letter · Decision Letter 2]

31 Aug 2020

Impact of the extension of a performance-based financing scheme to nutrition services in Burundi on malnutrition prevention and management among children below five: a cluster-randomized control trial

PONE-D-19-30259R2

Dear Dr. Korachais,

We’re pleased to inform you that your manuscript has been judged scientifically suitable for publication and will be formally accepted for publication once it meets all outstanding technical requirements.

Kind regards,

Seth Adu-Afarwuah

Academic Editor

PLOS ONE

Additional Editor Comments (optional):

Reviewers' comments:

Reviewer's Responses to Questions

**Comments to the Author**

1. If the authors have adequately addressed your comments raised in a previous round of review and you feel that this manuscript is now acceptable for publication, you may indicate that here to bypass the “Comments to the Author” section, enter your conflict of interest statement in the “Confidential to Editor” section, and submit your "Accept" recommendation.

Reviewer #3: All comments have been addressed

2. Is the manuscript technically sound, and do the data support the conclusions?

Reviewer #3: (No Response)

3. Has the statistical analysis been performed appropriately and rigorously? 

Reviewer #3: (No Response)

4. Have the authors made all data underlying the findings in their manuscript fully available?

Reviewer #3: (No Response)

5. Is the manuscript presented in an intelligible fashion and written in standard English?

Reviewer #3: (No Response)

6. Review Comments to the Author

Reviewer #3: (No Response)

7. PLOS authors have the option to publish the peer review history of their article (what does this mean?). If published, this will include your full peer review and any attached files.

Reviewer #3: No

---

## [Editor Report · Acceptance letter]

8 Sep 2020

PONE-D-19-30259R2 

Impact of the extension of a performance-based financing scheme to nutrition services in Burundi on malnutrition prevention and management among children below five: a cluster-randomized control trial 

Dear Dr. Korachais:

I'm pleased to inform you that your manuscript has been deemed suitable for publication in PLOS ONE. Congratulations! Your manuscript is now with our production department. 

Kind regards, 

on behalf of

Dr. Seth Adu-Afarwuah 

Academic Editor

PLOS ONE